# Fragility Fractures and Imminent Fracture Risk in the Spanish Population: A Retrospective Observational Cohort Study

**DOI:** 10.3390/jcm10051082

**Published:** 2021-03-05

**Authors:** Maria-José Montoya-García, Mercè Giner, Rodrigo Marcos, David García-Romero, Francisco-Jesús Olmo-Montes, Mª José Miranda, Blanca Hernández-Cruz, Miguel-Angel Colmenero, Mª Angeles Vázquez-Gámez

**Affiliations:** 1Departamento de Medicina, Universidad de Sevilla, Avda. Dr. Fedriani s/n, 41009 Sevilla, Spain; pmontoya@us.es (M.-J.M.-G.); mavazquez@us.es (M.A.V.-G.); 2Departamento de Citología e Histología Normal y Patológica, Universidad de Sevilla, Avda. Dr. Fedriani s/n, 41009 Sevilla, Spain; 3Orthopedic Surgery and Traumatology Service, Virgen Macarena University Hospital, Avda Sánchez Pizjuán s/n, 41009 Seville, Spain; rodri.marcos.rabanillo@gmail.com (R.M.); garciaromero5@hotmail.com (D.G.-R.); 4Servicio de Medicina Interna, HUV Macarena, Avda Sánchez Pizjuán s/n, 41009 Sevilla, Spain; franciscoj.olmo.sspa@juntadeandalucia.es (F.-J.O.-M.); m.j.mir@telefonica.net (M.J.M.); mangel.colmenero.sspa@juntadeandalucia.es (M.-A.C.); 5Rheumatology Service, Virgen Macarena University Hospital, Avda Sánchez Pizjuán s/n, 41009 Seville, Spain; blancahcruz@gmail.com

**Keywords:** osteoporosis, fragility fracture, fracture risk, imminent fracture risk

## Abstract

Fragility fractures constitute a major public health problem worldwide, causing important high morbidity and mortality rates. The aim was to present the epidemiology of fragility fractures and to assess the imminent risk of a subsequent fracture and mortality. This is a retrospective population-based cohort study (n = 1369) with a fragility fracture. We estimated the incidence rate of index fragility fractures and obtained information on the subsequent fractures and death during a follow-up of up to three years. We assessed the effect of age, sex, and skeletal site of index fracture as independent risk factors of further fractures and mortality. Incidence rate of index fragility fractures was 86.9/10,000 person-years, with highest rates for hip fractures in women aged ≥80 years. The risk of fracture was higher in subjects with a recent fracture (Relative Risk(RR), 1.80; *p* < 0.01). Higher age was an independent risk factor for further fracture events. Significant excess mortality was found in subjects aged ≥80 years and with a previous hip fracture (hazard ratio, 3.43 and 2.48, respectively). It is the first study in Spain to evaluate the incidence of major osteoporotic fractures, not only of the hip, and the rate of imminent fracture. Our results provide further evidence highlighting the need for early treatment.

## 1. Introduction

Fragility fractures caused by osteoporosis constitute a major public health problem worldwide. The annual costs attributable to fragility fractures in the European Union (EU) currently equate to €37 billion, and these numbers are expected to rise due to population aging [1]. Fragility fractures are an important cause of disability, morbidity, and mortality in the population [2]. Such massive burden highlights the importance of risk assessment of fragility fractures and the need to adapt prevention strategies to individual risk patterns.

A large number of risk factors for fragility fracture have been identified [3]. Among them, a previous fragility fracture has been generally recognized as an assessment criteria of fracture risk in osteoporosis, regardless of bone mineral density (BMD) [4]. Although it is generally recognized that the risk of fracture increases throughout life with a previous fracture, a recent osteoporotic fracture increases even more the risk of an imminent fracture [5,6,7], and the magnitude of this risk and the contribution of other clinical risk factors still demand further research. Previous evidence suggests that the risk is not constant but rather fluctuates over time, being greatest within the first few years of the initial fracture [5,8,9,10,11]. The predictive importance of imminent (12 to 24 months) fracture risk is widely accepted. However, the significance of the early years after an index fracture need to be further explored. After a first fragility fracture, the skeletal location of the index fracture may also influence the magnitude of imminent fracture risk. However, only a few studies have measured its effect on the level of further fracture risk, and conclusions do not hold for solid generalization. [12,13,14]. Older age is also a well-defined clinical risk factor for fractures [15]. However, there is controversy between studies over its contribution on the risk of further fracture events. Some authors have observed a marked increase by age in the risk of a second major fragility fracture [11,16], whereas the association between age and subsequent fracture was not confirmed by other reports [3,17]. As with age, the predictive value of sex in the risk of further fracture events is also controversial. Originally, women were considered at higher risk of both initial and subsequent fracture and, indeed, some authors confirmed this hypothesis [11,16]. However, other studies have not reported differences in the risk of subsequent fracture events between men and women [18,19].

The primary objective of the present study was to explore the epidemiology of fragility fractures in southern Spain by using data from a population-based cohort of women and men, aged 50 or older, admitted to the emergency unit at Virgen Macarena University Hospital. Our study also provides an estimation of the incidence rate of subsequent fractures over 1–3 years following an index fracture. As secondary objectives, the study evaluated mortality over three years following an index fracture, analyzing the role of subsequent fractures as an independent risk factor.

## 2. Materials and Methods

This study was designed as a retrospective observational population-based cohort study involving men and women aged ≥50 years with an index fragility fracture (caused by an injury that would be insufficient to fracture normal bone; the result of a/or resistance to bone torsion [20]) occurring between 1 January 2014 and 31 December 2014.

Eligible study participants were followed until 31 December 2016, to obtain information on the outcomes: subsequent fragility fractures and deaths events. Age, sex, time from index fracture, and skeletal site were assessed as potential independent risk factors. Patients were identified using the emergency unit’s medical records at Virgen Macarena University Hospital in Spain. Virgen Macarena University Hospital is a public tertiary hospital and its Emergency Unit serves up to 481,879 inhabitants, of whom 157,428 subjects are aged ≥ 50 years, within the healthcare area of North Seville. Furthermore, this is the single reference hospital of the healthcare area including emergency care attention. General medical records (Diraya) were also used to collect any relevant information on the study outcomes including demographic information, index fracture type, and causality as well as any relevant radiological confirmatory findings. Eligible incidental fragility fracture locations included axial (hip, pelvis, dorsal, and lumbar vertebrae) as well as appendicular bones (proximal humerus and wrist), according to the International Classification of Diseases, ICD-9 codes (Appendix A). Diagnosis was based on symptoms but must have included a radiologic confirmation of the fracture. Non-clinical radiographic vertebral fractures as well as other pathological or traumatic fractures were excluded. The identification of subsequent fractures required a similar main diagnosis of fracture. To distinguish subsequent fractures from previous events recorded at follow-up visits and/or patient history, the following criteria were applied: (1) Fractures in the same skeletal site of index fracture were only captured if a minimum of four months had elapsed since the index fracture; (2) hip fractures were only captured if an inpatient hospital admission was required; (3) all medical visits identified as follow-up examination of a previous fracture were excluded as further fracture events; (4) patients who died following a fracture were captured as having both outcomes; and (5) if the index fracture involved more than one skeletal sites, to avoid double counting, the fracture was assigned to the site of highest severity. Time at risk for subsequent fracture events began the day after the date of the index fracture and continued until outcome occurrence, either fracture or patient death.

We estimated the incidence rate over 12 months of (index) fragility fractures in the general population aged ≥ 50 years (based on a total estimated population of 157,428 individuals aged 50 or over served in Virgen Macarena Hospital catchment area on 1 January 2014). Then, we estimated the incidence rate of further fracture events during the study period among those who had a previous fragility fracture in 2014. Time at risk for subsequent fracture events began the day after the date of the index fracture and continued until outcome occurrence, either fracture or patient death. Fracture incidence rates per 10,000 person-years were calculated by age group, sex, and fracture type. The 95% confidence intervals (CIs) were calculated assuming a Poisson distribution. The excess risk of further fractures was compared to the general population using a Poisson regression model including age, sex, and location of previous fracture as covariates. Nelson–Aalen cumulative hazard estimates were plotted to analyze the time to a subsequent fracture event [21]. The risk of subsequent fracture was analyzed using a Cox proportional hazards regression model, as well as the model proposed by Fine and Gray [22], using death as a competing risk. All-cause mortality was also analyzed using the Cox hazard model. Furthermore, we included a time-dependent variable in this model to estimate the risk of all-cause death associated with the occurrence of subsequent fracture events during the study period. Estimates with *p* values < 0.05 were considered statistically significant. No imputation of missing data was necessary. All statistical analyses were performed using Stata software (STATA Corp., College Station, TX, USA).

## 3. Results

### 3.1. Patient Baseline Characteristics

Among a total population of 157,428 Caucasian individuals aged 50 or over served by Virgen Macarena University Hospital, 1068 women and 301 men (3.5 female/male ratio), with mean of age 75.1 and 72.1 years, respectively, registered eligible index fragility fractures in 2014 and were included in the analyses. Only 14 subjects were excluded due to miscoding of fractures (3/14) or traumatic (8/14) or pathological fractures (3/14), (Figure 1). The most frequent index fracture site in women was wrist (405 [37.9%]), whereas in men hip fractures were the most common (111 [36.9%]). Mean duration of follow-up was 2.3 years for all subjects, 2.2 years in males and 2.3 in females.

### 3.2. Incidence Rate of Index Fracture in the General Population

An overall incidence rate of 86.9 fractures/10,000 person-years was found in the general population aged ≥ 50 years (Table 1). The frequency of fragility fractures was significantly higher among women as compared to men: 123.9 (CI 95% 116.6–131.6) versus 42.3 (CI95% 37.7–47.4), respectively. Also, marked increase in the frequency was also observed with increasing age. The highest overall frequency was observed for wrist fractures. Despite this, hip fractures were the most frequent among women and men aged ≥ 80 years. On the other hand, the skeletal sites with the lowest incidence rate of fragility fractures were the pelvis and the spine.

### 3.3. Incidence of Subsequent Fractures

The frequency of clinical subsequent fracture events was 318.2/10,000 person-years. Incidence rate of subsequent fractures was higher in women than in men. However, the overall differences did not reach statistical significance (Table 2). By contrast, the frequency of clinical fracture events increased with age, with markedly higher incidences rates in men and women aged 70 years or older (Table 2). Overall, no significant differences in the frequency of subsequent fracture events were observed by index fracture type. However, a slight trend was observed toward increased incidence among subjects with pelvis and hip index fractures (Table 2). The rate of further fracture events was highest within the ≥80 years’ age group, in women with a previous fracture in the pelvis and men with a previous wrist index fracture. The most frequent skeletal locations of further fractures were hip and wrist. The incidence rate of further fracture events during the first year after the index fragility fracture was not higher than the rates observed during the following second and third years of follow-up, regardless of sex and age (Figure 2). Similarly, no marked differences among follow-up periods were observed by site of index fracture. However, a slight trend was observed for wrist fractures (Appendix A).

### 3.4. Risk Factors of Subsequent Clinical Fracture

Overall, the incidence rate of fractures was higher for subjects with a previous index fragility fracture at any site compared with the general population (relative risk [RR] 1.80, *p* < 0.01) (Figure 3 and Appendix A). Independent risk factors for subsequent fracture, as identified by multivariate analysis using Cox as well as Fine and Gray regression models, Higher age (≥70 years) was an independent risk factor for further fracture events, with a ≥1.5 increase in hazard risk (HR) observed for each decade from 60 years of age (Table 3). Multivariate analysis using Fine and Gray model revealed an increased risk of subsequent fractures in women. No effect of index fracture site on the level of risk of further fractures was observed (Table 3).

### 3.5. Risk of Death Following Initial Fracture

A total of 120 deaths occurred during the study follow-up period in patients aged ≥50 years with an index fragility fracture, with overall mortality rates reaching 37.20/1000 person-years (29.09 and 67.42 per 1000 person-years in women and men, respectively). Mortality rates were higher among men as compared to women (HR, 0.41; *p* < 0.01) (Table 4). Age was the strongest determinant of mortality with significant excess risk for subjects aged 80 or older (HR, 3.43 *p* < 0.01). Mortality risk was also significantly higher among patients presenting an index fragility fracture in the hip (HR, 2.48; *p* < 0.01). Lower mortality rates were observed in subjects with index fractures located in peripheral bone positions (wrist and proximal humerus). Mortality risk also increased after a subsequent fracture occurred, although this association did not reach statistical significance (HR, 2.14; *p* = 0.06) (Table 4).

## 4. Discussion

This study presents the first report of the incidence rate of index of major fragility fractures and the risk of imminent fractures in a Spanish cohort of 1369 subjects (1068 women and 301 men) aged ≥50 years, by age group and fracture site. Our study confirmed markedly higher rates in women, as well as an age-related increase in the risk, with highest frequency rates found in women aged ≥80 years. The prevalence of osteoporosis and osteoporosis fracture rates is higher in women compared to men. This is due to differences in BMD, bone size, bone geometry, and bone strength [23,24]. Estrogen deprivation after menopause is a major contributing factor, which could be the reason for the observed gender-related differences [25]. Age, on the other hand, is a well-studied risk factor of index fragility fractures, contributing to risk independently of bone mass density [26]. Previous information on the incidence rate of fragility fractures in Spain is scarce. As a first approach, using the Q-FRACTURE tool, González López-Valcárcel et al. [27] estimated a level of risk ranging from 1.8–21.5% in women and 0.7–10.8% in men. According to our findings, the frequency of osteoporotic fractures in Spain may be sensitively higher than reported. Conversely, our numbers underestimate the crude rates published by the International Osteoporosis Foundation for Spain as well as for other EU countries [28]. This discrepancy may be partly explained by the exclusion of non-clinical vertebral fractures and other fracture sites less commonly associated with osteoporosis. Overall, the most frequent index fracture type was wrist. However, the rate of hip fractures exceeds that of wrist in older aged groups. Similar age-related trends in the frequency of fragility fractures were reported previously [29].

Estimated incidence rate of subsequent fractures was 318.2/10,000 person-years in all subjects during the three years that followed index fracture (i.e., 3.2% of patients with a previous fracture experienced a new fracture every year). To date, no other studies have been published that measure the risk of imminent fractures after a sentinel fracture in Spain. Only Azagra et al. have published 10-year fracture data in a population cohort in Catalonia that presented clinical risk factors for osteoporotic fractures, with the aim of validating the Frax tool in the Spanish population [30].

Overall, the incidence of fracture was higher for subjects with a previous index fragility fracture at any site, compared with the frequency in the general population aged 50 or older (RR, 1.80). According to our findings, Kanis et al. [17] observed that, for any type of previous fracture, the RR of any further fracture ranged from 1.83 to 2.03 depending upon age. The effect of gender as a predictor of the risk was only significant when the analysis considered death as a competing risk. Previous reports observed similar risks in men and women, except among subjects over 85 years of age [31]. Our data also, however, proved the well-known independent effect of aging on the risk of further fracture events, [11,14], with significant differences in the HR among older age (≥70 years). They also found a marked age-related increase in the risk but did not observe any differences in the risk among women and men.

Noticeable differences in the frequency of subsequent fractures were found depending upon site, with highest rates found in subjects with a previous pelvis or hip fracture. Our findings, however, could not confirm the effect of the skeletal site on the risk of further fractures. Previous reports on the associations between prior and subsequent fractures are not consistent [7]. The time that follows initial fracture is key with regard to the risk and prevention of subsequent fracture events. Several previous studies report that the highest risk of further fracture events occurs within the first year after the index fracture [8,9,10,11] and that the incidence decreases thereafter. In the current study, however, the incidence of subsequent fracture events during the first year after the index fracture was not higher than the rates observed during the following second and third years of follow-up, regardless of sex and age. Despite this, a slight trend was observed for wrist index fractures. Authors reporting higher levels of risk during the first year after the index fracture have assessed longer timescales than our study [9]. Like in our study, Banefelt et al. [11] focused only in the early years following the index fracture and found the highest incidence during the second year (12%), rather than the first (7.1%).

Overall mortality rate reached 37.20/1000 person-years, which showed an up-to-2-fold excess mortality due to osteoporotic fractures among the younger-aged groups [32]. The risk of death was significantly higher in patients with a previous hip index fracture (HR, 2.48; *p* < 0.01), with an estimated rate of 73.50/1000 person-years. No significant excess mortality was found for index fractures in other skeletal locations or for the event of subsequent fractures (*p* = 0.06). As expected, our data also confirmed a higher age-specific death risk (*p* < 0.01, in subjects aged ≥80 years) as well as lower death risk in women (*p* < 0.01). In fact, 58.3% of all deaths occurred in patients with a previous hip index fracture and a mean age of 83.3 years. The observed death rate after a hip fracture was sensitively lower than previously reported [33,34], which could be explained by the longer observation period of this study, as mortality is highest during the six months that follow the event [35].

The main limitation of our study is the lack of data from other clinical factors that could have contributed to in-depth understanding of the risk of subsequent fractures (bone mass density, previous record of falls, history of prior fractures, use of drugs affecting bone metabolism). One of the strengths of our study is that, as opposed to database studies, we manually reviewed clinical records of all 1369 cases to confirm eligibility as well as outcomes’ information.

## 5. Conclusions

In summary, this report provides information on the magnitude and consequences of fragility fractures in the as-yet unexplored Spanish population aged ≥50 years, involving both genders, as well as major skeletal sites associated with osteoporosis, not limited to hip. This is the first study to report on the incidence rate of imminent fractures after an osteoporotic sentinel fracture in Spain, with age, sex, and skeletal location of the index fractures as possible risk factors. Our results support the increased risk of imminent fracture after a recent fracture and provide key elements for early identification of risk and the application of targeted strategies aimed at preventing future fractures and mortality, such as the interventions recommended by the Fracture Liaison Services [36].

## Figures and Tables

**Figure 1 jcm-10-01082-f001:**
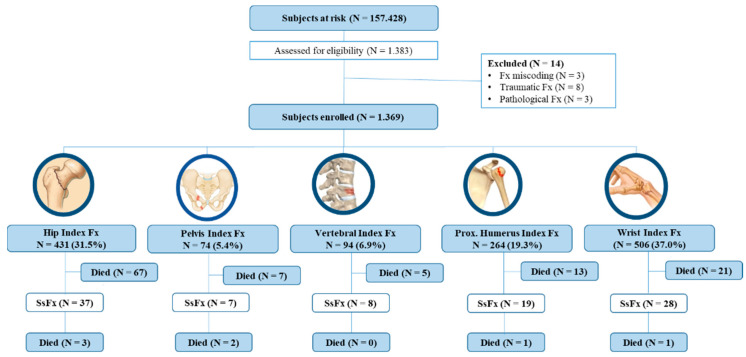
Flow diagram of study participants based on the Strobe statement. A total of 1383 subjects were assessed for eligibility, among a study population of 157,428 inhabitants. Fracture miscoding or traumatic or pathological causes of fractures were the causes for the exclusion of 14 subjects. Subjects were enrolled from January 2014 to December 2014 and followed up until December 2016. Mean duration of follow-up was 2.3 years for all subjects, 2.2 years in males and 2.3 in females. The most frequent types of index fracture were wrist (N = 506, 37.0%) and hip (N = 431, 31.5%), followed closely by proximal humerus (N = 264, 19.3%). Overall, 99 subjects registered a subsequent fracture, with 120 events of death occurring throughout follow-up. Fx: fracture. SsFx: subsequent fracture.

**Figure 2 jcm-10-01082-f002:**
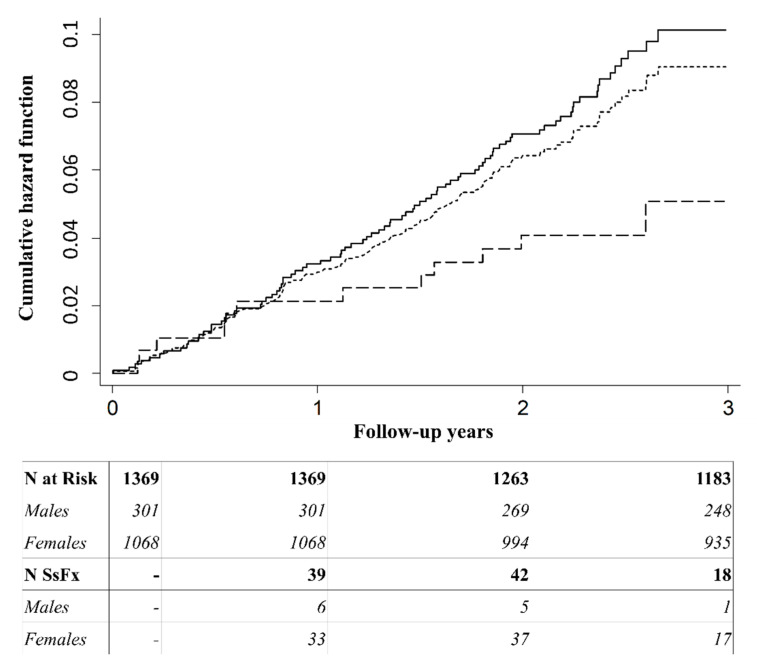
Nelson–Aalen cumulative hazard of subsequent fracture events after an index fracture in men and women aged ≥50 years over a period of three years of follow-up. Cumulative risk of subsequent fracture increased over the years following initial fracture. However, no significant differences were found in the incidence rate of subsequent fractures over 1, 2, and 3 years, regardless of sex and age. Dotted line represents the cumulative incidence of subsequent fractures in all subjects. Solid line represents the cumulative incidence of subsequent fracture events in females. Dashed line represents the cumulative incidence of subsequent fracture events in males. SsFx: subsequent fracture. The Y-axis represents the cumulative hazard function. The X-axis represents follow-up years after the first fracture.

**Figure 3 jcm-10-01082-f003:**
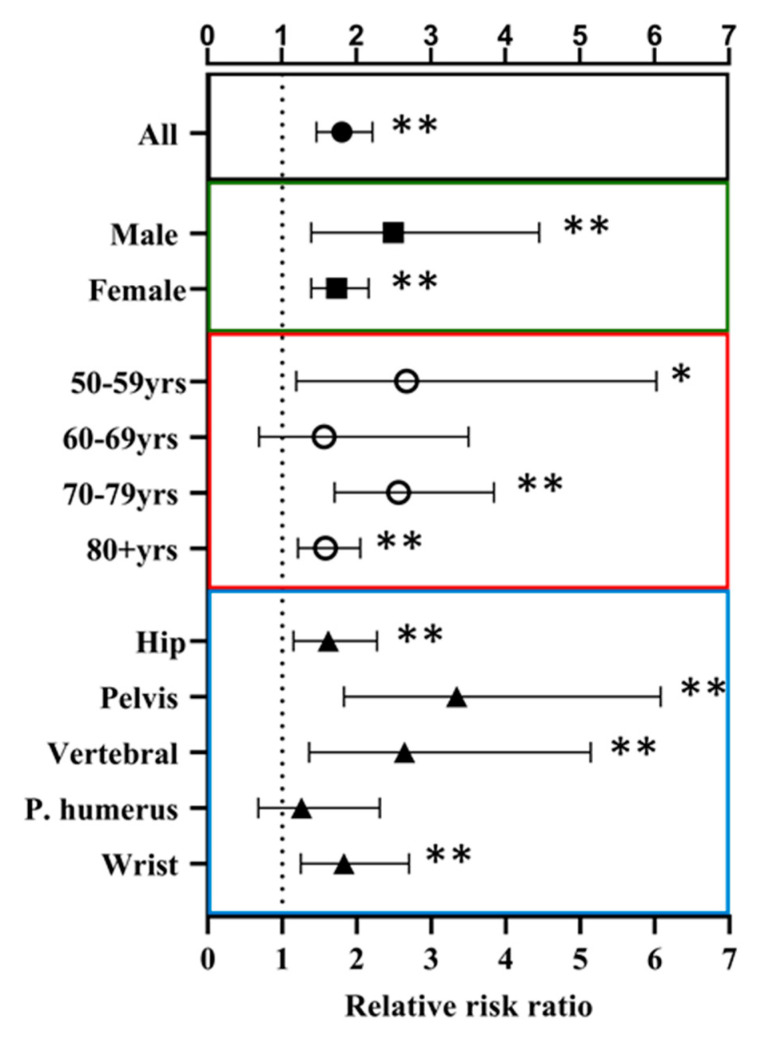
Relative risk of subsequent fractures among subjects with a previous fracture compared with the general population by sex (green box and square), age group (red box and circle), and index fracture site (blue box and triangle). Adjusted rate ratio estimated using Poisson regression models that included age, sex, and index fracture site as covariates. Corresponding data are presented in Appendix A. * *p* < 0.05, ** *p* < 0.01.

**Table 1 jcm-10-01082-t001:** Incidence rate of index fracture in the general population aged ≥50 years, by index fracture site, sex, and age group. The total population values for males and females are marked in bold. The shading indicates that they are total values of the population, to differentiate them from the rest that are separated by gender and age.

			Index Fracture Site				
	Age *	All Sites	Hip	Pelvis	Vertebral	Prox. Humerus	Wrist
**All subjects**	**50+ years**	**86.9 (82.4–91.6)**	**27.5 (25.0–30.2)**	**4.6 (3.6–5.8)**	**6.0 (4.8–7.3)**	**16.8 (14.8–18.9)**	**32.1 (29.4–35.0)**
**Males**	**All males**	**42.3 (37.7–47.4)**	**15.6 (12.8–18.8)**	**1.8 (0.0–3.1)**	**2.4 (1.4–3.8)**	**8.3 (6.3–10.7)**	**14.2 (11.6–17.3)**
	**50–59 years**	23.4 (18.3–29.5)	2.3 (0.9–4.8)	0.0 (NA)	1.3 (0.4–3.4)	7.3 (4.5–11.0)	12.5 (8.9–17.2)
	**60–69 years**	24.7 (18.5–32.3)	4.7 (2.2–8.6)	0.9 (0.1–3.4)	0.9 (0.1–3.4)	6.1 (3.2–10.4)	12.1 (7.9–17.8)
	**70–79 years**	56.4 (44.1–71.0)	18.8 (12.1–28.0)	2.4 (0.5–6.9)	3.1 (0.9–8.0)	11.8 (6.6–19.4)	20.4 (13.3–29.8)
	**80+ years**	157.7 (129.0–190.9)	105.1 (82.0–132.8)	12.0 (5.2–23.7)	10.5 (4.2–21.7)	13.5 (6.2–25.7	16.5 (8.25–29.6)
**Females**	**All females**	**123.9 (116.6–131.6)**	**37.1 (33.2–41.4)**	**7.1 (5.4–9.1)**	**8.9 (7.1–11.2)**	**23.8 (20.6–27.3)**	**47.0 (42.5–51.8)**
	**50–59 years**	48.3 (41.00–6.5)	3.7 (1.9–6.5)	0.6 (0.1 -2.3)	4.4 (2.4–7.3)	12.2 (8.6–16.6)	27.4 (22.0–33.8)
	**60–69 years**	77.9 (67.1–89.9)	5.8 (3.2–9.8)	2.5 (0.9–5.4)	2.9 (1.2–6.0)	19.6 (14.4–26.0)	47.1 (38.8–56.6)
	**70–79 years**	150.2 (132.4–169.7)	38.0 (29.3–48.4)	8.2 (4.5–13.7)	13.4 (8.5–20.2)	31.6 (23.7–41.2)	59.0 (48.1–71.7)
	**80+ years**	359.5 (327.7–393.6)	175.5 (153.5–199.8)	29.9 (21.3–40.9)	25.3 (17.4–35.5)	49.8 (38.5–63.5)	79.0 (64.4–95.8)

Data are incidence rate per 10,000 person-years (95% confidence interval). * Age is described as of index date.

**Table 2 jcm-10-01082-t002:** Incidence rate of subsequent fractures (any site) among patients with an index fracture, by index fracture site, sex, and age group. The total population values for males and females are marked in bold. The shading indicates that they are total values of the population, to differentiate them from the rest that are separated by gender and age.

				Index Fracture Site			
	Age *	All Sites	Hip	Pelvis	Vertebral	Prox. Humerus	Wrist
**All subjects**	**50+ years**	**318.2 (261.3–387.5)**	**406.8 (294.7–561.4)**	**425.8 (203.1–893.1)**	**364.5 (182.3–728.9)**	**306.0 (195.2–479.8)**	**234.0 (161.6–338.9)**
**Males**	**All males**	**180.4 (93.2–315.1)**	**228.8 (74.3–534.0)**	**0.0 (NA)**	**504.4 (61.1–1821.9)**	**69.7 (1.8–388.5)**	**171.4 (46.7–438.8)**
	**50–59 years**	56.4 (1.4–314.2)	0.0 (NA)	0.0 (NA)	1014.3 (25.7–5651.3)	0.0 (NA)	0.0 (NA)
	**60–69 years**	0.0 (NA)	0.0 (NA)	0.0 (NA)	0.0 (NA)	0.0 (NA)	0.0 (NA)
	**70–79 years**	259.4 (70.7–664.1)	464.9 (56.3–1679.4)	0.0 (NA)	0.0 (NA)	0.0 (NA)	359.8 (43.6–1299.5)
	**80+ years**	333.0 (133.9–686.2)	218.9 (45.2–639.8)	0.0 (NA)	665.4 (16.9–3707.5)	473.4 (12.0–2637.8)	1048.8 (127.0–3788.7)
**Females**	**All females**	**355.7 (284.9–438.8)**	**463.0 (316.7–653.6)**	**521.8 (209.8–1075.1)**	**333.7 (122.4–726.2)**	**377.0 (223.4–595.8)**	**249.1 (159.6–370.7)**
	**50–59 years**	131.2 (42.6–306.3)	0.0 (NA)	0.0 (NA)	275.8 (7.0–1536.8)	107.0 (2.7–596.1)	137.7 (28.4–402.4)
	**60–69 years**	131.5 (48.3–286.2)	0.0 (NA)	0.0 (NA)	0.0 (NA)	0.0 (NA)	217.8 (79.9–474.0)
	**70–79 years**	352.4 (218.1–538.6)	414.11 (152.0–901.3)	0.0 (NA)	571.2 (117.8–1669.2)	658.4 (284.3–1297.3)	163.5 (44.5–418.6)
	**80+ years**	543.2 (409.2–707.0)	540.6 (353.1–792.1)	852.6 (342.8–1756.7)	262.7 (31.8–948.8)	607.8 (277.9–1153.8)	488.3 (243.8–873.7)

Data are incidence rate per 10,000 person-years (95% confidence interval). * Age is described as of index date.

**Table 3 jcm-10-01082-t003:** Hazard ratio (Cox) and subhazard ratio (Fine and Gray) of subsequent fracture events associated to sex, age, and index fracture site.

Risk Factor	HR ^†^	95% CI	*p* > |z|	SHR ^‡^	95% CI	*p* > |z|
**Sex**						
Male *	1			1		
Female	1.73	(0.94–3.17)	0.08	1.87	(1.01–3.46)	0.05
**Age group**						
50–59 years *	1			1		
60–69 years	0.92	(0.30–2.86)	0.88	0.91	(0.30–2.78)	0.87
70–79 years	2.94	(1.20–7.22)	0.02	2.88	(1.18–7.05)	0.02
80+ years	4.41	(1.85–10.51)	<0.01	4.15	(1.74–9.89)	<0.01
**Index fracture site**						
Hip	1.05	(0.62–1.77)	0.85	0.98	(0.58–1.66)	0.94
Pelvis	1.14	(0.49–2.64)	0.77	1.12	(0.49–2.57)	0.79
Vertebral	1.14	(0.52–2.53)	0.74	1.14	(0.51–2.53)	0.75
Prox. humerus	1.20	(0.67–2.15)	0.54	1.22	(0.68–2.18)	0.51
Wrist *	1			1		
**Index fracture type ^+^**						
Appendicular	1			1		
Central	1	(0.66–1.53)	0.99	0.95	(0.61–1.46)	0.80

* Baseline category; ^†^ hazard ratio (HR) and 95% confidence interval (CI) estimates adjusted for all variables in the table using a Cox proportional hazards model. ^‡^ Subhazard ratio (SHR) and 95% confidence interval (CI) estimates adjusted for all variables in the table using a Fine and Gray competing risks model using death as competing risk. ^+^ Appendicular fractures: wrist and proximal humerus. Central fractures: vertebral, pelvis, and hip fractures.

**Table 4 jcm-10-01082-t004:** Mortality hazard ratio associated to sex, age, type of index fracture, and presence of subsequent fracture events.

Risk Factor	MR × 1000 PY ^‡^	HR ^†^	95% CI	*p* > |z|
**Sex**				
Male *	67.42	1		
Female	29.09	0.41	(0.28–0.60)	<0.01
**Age group**				
50–59years *	12.42	1		
60–69years	17.04	1.43	(0.54–3.77)	0.47
70–79years	29.30	2.00	(0.84–4.73)	0.12
80+years	62.00	3.43	(1.51–7.77)	<0.01
**Index fracture site**				
Hip	73.50	2.48	(1.47–4.19)	<0.01
Pelvis	52.60	1.95	(0.88–4.35)	0.10
Vertebral *	22.07	1.01	(0.38–2.68)	0.99
Proximal humerus	21.83	1.13	(0.58–2.21)	0.73
Wrist	17.82	1		
**Presence of SsFX**				
no SsFX *	36.32	1		
SsFX	60.89	2.14	(0.97–4.70)	0.06

* Baseline category. ^†^ Mortality hazard ratio (HR) and 95% confidence interval (CI) estimates adjusted for all variables in the table using a Cox proportional hazards model. ^‡^ Mortality rate (MR) per 1000 person-years (PY). SsFx: subsequent fracture.

## Data Availability

The data presented in this study are available on request from the corresponding author. The data are not publicly available due to privacy.

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
