# Peer review of "Fragility Fractures and Imminent Fracture Risk in the Spanish Population: A Retrospective Observational Cohort Study"

_jcm, 2021, doi:10.3390/jcm10051082_

Round 1

Reviewer 1 Report

This study presents a report on the incidence rate and risk of fractures in a Spanish cohort of 1369 subjects. The study is very well conducted as it provides early identification of risk with the aim of preventing future fractures from happening.

Minor comments:

Line 117: Please cite any previous study that has used NELSON-ALEN cumulative hazard to analyse time to a fracture event.

Figure 2: Legend on the Y and X-axis missing

figure 3; Legend on X-axis missing

Line 257-258: Please explain scientifically why prevalences of Osteoporosis is higher in woman than in man. In 1-2 sentences.

Line 273: The author claims there are no other studies that measure the risk of further fractures in Spain. However, Azagra et.al 2011 has published a similar study. Please explain how your study is different from Azagra et.al 2011. (https://link.springer.com/article/10.1186/1471-2474-12-30)

Line 320: same as above.

Reviewer 2 Report

Thank you for this interesting article which addresses an important topic which will affect Orthopedic Traumatology for the next years. Following you find some adjustments which needs to be addressed.

It would be necessary to address the introduction where you state to reach out for epidemiology of fragility fractures. This is already well described in literature and would no necessarily add to the literature itself. In Line 312 you state missing information “bone mass  density, previous record of falls, history of prior fractures, use of drugs affecting bone metabolism”which would be of great interest and importance yet bone mass density is one crucial factor to call a fracture “fragility fracture”. Did you scan patients for BMD when fractures occurred and is there a routine protocol in your clinic? Sure there will be why I would recommend to include this in the “Materials and Methods” Section.

I would not call fractures in 51 yr old patients “fragility” fractures. To date data implies an age of 65 and older. In case of osteoporosis the fragility fracture can also occur in younger patients why it would be necessary to include BMD data in your article.

Otherwise please address the age profile in the article

Furthermore please add one sentence for the readers why imminent fractures are addressed. It is worthy to “overmention” this to gain awareness on this topic.

Line 20: rephrase (important morbidity / mortality) “high rates” etc

Line 40: First use of “EU” --> European Union

Line 40: Reference at the End of the Sentence

Line 72: “in the southern spain”

Line 93: not sure if the Colles Fracture itself needs to be called here. Wrist fractures overall should be ok.

Line 113: You calculated fractures / 100.000 inhabitants. In the following article you give numbers /10. 000. Please rephrase or adapt Results.

Results: The results section should be without reference if not necessarily needed.

Line 131: Only Caucasian patients or overall inhabitant counts? i

Line 133: 75,1 nd 72.1 --> stick to 72.1 rather than 75,1 (no comma)

Line 155: rather “to” --> women as compared with men

Line155: how much higher was the rate since you state “significantly”.

Line 253: wrist fractures are obviously no “major”fragility fracture. Rephrase.

Round 2

Reviewer 2 Report

Thanks again for the interesting work and the now revised version.